# Statistical Evaluation of Quantities Measured in the Detection of Soil Air Pollution of the Environmental Burden

Erika Skvarekova [1],*, Marcela Tausova [1], Andrea Senova [1], Gabriel Wittenberger [1] and Jana Novakova [2]

1   Institute of Earth's Resources, Department of Montaneous Sciences, Faculty of Mining, Ecology, Process Control and Geotechnologies, Technical University of Kosice, Park Komenskeho 19, 04001 Kosice, Slovakia; marcela.tausova@tuke.sk (M.T.); andrea.senova@tuke.sk (A.S.); gabriel.wittenberger@tuke.sk (G.W.)
2   U.S. Steel Kosice-Labortest, 04454 Kosice, Slovakia; novjane@centrum.sk
*   Correspondence: erika.skvarekova@tuke.sk; Tel.: +421-55-6023148

**Abstract:** The article highlights the investigation of the relationships between measured quantities during the atmospheric geochemical survey of contaminated soil and the environmental burden of the industrial establishment in eastern Slovakia. Statistical data processing was undertaken from the measured values of pollutants. The basic statistical characteristics of the monitored indicators were defined here. With the help of regressive and correlative analysis, dependency was confirmed between examined values, further expressed by a mathematical relationship. We analysed variability of the measured variables due to the influence of changed input quantities by the non-parametric Wilcox test. The statistical data processing helps us to identify the dependency between the measured values and improves valorization of the pollution of a given environmental burden. This was due to the handling of organic pollutants and the production of basic organic and inorganic chemicals stated for other industries. Chemical analysis of soil air helps us to determine the extent and amount of soil contamination by pollutants. Individual pollutants have their own characteristic properties and their negative effects on biota, the environment and humans are different.

**Keywords:** soil; soil air; statistical data processing; soil contamination; atmospheric geochemical survey; environmental burden





## 1. Introduction

Soil pollution caused by human activity can pose a serious risk to human health, the environment, groundwater and soil. Diagnosis of ecosystem damage at an early stage is an important task in terms of preventive measures against toxic impacts [1,2].

Gaseous compounds found in the porous environment of soils and rock formations are referred to as soil air. The presence of organic compounds in the gaseous phase may point to pollution beneath the surface environment. The presence and movement of these substances can be detected by using a technique called a soil vapor survey (SVS), which is based on soil air analysis.

According to the method of SVE of soil contaminated with benzene, an extraction device was designed that can play a role in improving remediation efficiency and saving remediation costs [3].

Gaseous compounds located in porous soil environments and rock formation could be contaminated with organic substances. Soil gas surveys are used as a tool for detecting contaminants in soil.

This article processes the identified environmental pollution burden of an industrial establishment in eastern Slovakia according to project [4]. The environmental burden occurred due to the industrial production and handling with polychlorinated biphenyls PCBs and also due to production of basic organic and inorganic chemicals intended for other industries.

Article [5] presented measured polychlorinated biphenyl concentrations in surface soil samples (0−20 cm) across China in 2005. The article evaluated measured values of organically polluting substances ($CH_4$, total hydrocarbons, volatile substances) which are used for effective evaluation of the rock environment quality. It also used a method where air from soil was extracted via a modern multifunctioning analyser, which helped with the detection of pollution.

Statistical processing of data was supported by individual dependencies of quantities. These showed us the mutual interaction between evaluated parameters (for example: if the pollution of $CH_4$ is growing, then $CO_2$ is higher as well, which proves dependency between these two parameters).

This method of extraction of air from the soil is a first method used in a geological research survey. To the best of our knowledge, this method is rarely published and that is why we claim that publishing of our outcomes can support knowledge about this subject in the scientific community.

One serious problem the environment is facing, is the presence of potentially hazardous persistent organic pollutants [6,7]. During the detection of contamination of the environmental burden in the project [4], soil air samples from the soil of the discharge canal surroundings were analysed for the determination of hydrocarbon soil contamination and for the monitoring of toxic gases and petroleum substances through the analyser atmospheric geochemical survey method.

Firms in the USA and European Union (EU) use integrated passive soil gas surveys utilize versatile technologies that have been demonstrated to provide the most accurate data on sites where organically polluting substances are of concern.

This analyser can also be used to monitor the migration of gases from the reaction zone towards the surface to detect leakages and to assess chemical hazards on the surface when identifying and quantifying chemical hazards during a field experiment with underground coal gasification [8,9]. It was used to measure $CO_2$ and $O_2$ levels in the air from soil on a pile of waste ash from shale ash in Kvarntorp, Sweden, with an increased concentration of uranium [10]. The temperature and development of gases in the core of composting material were also monitored by analyser every day throughout the whole composting process [11]. The analyser was used while measuring the methane content during methane production of two synthetic organic fractions of municipal solid waste with different lignocellulose [12,13]. Volatile organic compounds (VOCs) were monitored with the analyser at a site mainly contaminated with psychopharmaceuticals and monoaromatic hydrocarbons such as benzene, toluene and chlorobenzene [14]. These applications show the diversity of analyser applications in practice. The data obtained can then be used to create statistical data processing of individual environmental burdens.

## 2. Materials and Methods

For our research we have chosen a company whose activity consists of chemical production, which made explosives and intermediate products intended for military and civilian purposes. Among the first products of the chemical process were formaldehyde and urotropine. Later, the manufacturing program was extended to the production of polychlorinated biphenyls. The total production of the company until 2016 was about 21,000 tons aimed mainly for eastern European countries. It produced also a mixture of polychlorinated biphenyls under the commercial title Delor, Hydelor and Deloterm. Later it produced other fertilizers and other goods such as, cyclohexane, cyclohexanon and cyclohexanol (rubber chemicals).

The task of the project carried out was to detect organic pollution of the sewage line, into which the company has for decades discharged chemically contaminated liquid waste from chemical treatment.

According to the project [4], which was focused on the geological survey of the company mentioned, the extent of the environmental burden was detected. From the

extent of the environmental burden we further employed the following procedures for atmospheric geochemical measurements in the field:

- During drilling work, basic atmospheric geochemical measurements were carried out according to a Standard STN 01 5113-Gas sampling [15].
- Due to atmospheric geochemical measurements we documented not only the soluble, sorbed, but also the gaseous component of contaminants.
- A detailed report on soil air measurement was prepared, which also included outputs and graphs from individual measurements.

Atmospheric geochemical measurements served to provide us with information on the evaporation of contaminants into the soil air in the aerating zone. The volatile petroleum hydrocarbon (TP) content of the soil air was measured by an IR analyser and also by a photo ionization detector (PID). In addition, other relevant soil air parameters such as $O_2$ percentage, $CO_2$ and temperature were also measured.

The analyser carried out 94 soil air measurements at depths of 2 and 4 m using atmospheric geochemical surveys. The parameters measured were $CH_4$, $CO_2$, TP (hydrocarbons overall), PID (volatile organic components of petroleum substances VOC, BTEX as a group indicator for benzene, toluene, ethylbenzene and xylene), $O_2$, vapor temperature.

Article [16] outlined ground-based determinative methods for oil VOCs, the interaction between oil VOCs and soil, and information on how they diffuse from underground into atmosphere.

Soil air extraction was carried out by means of analyser pumps and with sorption tubes, selected (according to the results of atmospheric geochemical measurements) mapping, hydrogeological wells and hand-drilled wells. We carried out 147 soil air extractions. All samples taken were transported to the appropriate chemical laboratory within 24 h. All samples taken were in collaboration with an accredited laboratory which carried out the analyses.

The Ecoprobe 5 gas analyser was used for the aforementioned experimental measurements. The analyser is a highly sensitive portable multifunctional gas analyser for soil contamination surveys in situ. It is a device with optimal performance and flexibility for effective surveys of organic pollution contamination and monitoring of volatile organic compounds, i.e., light non-aqueous phase liquid benzene, toluene, ethylbenzene and xylene, in soil (Figure 1) [17].

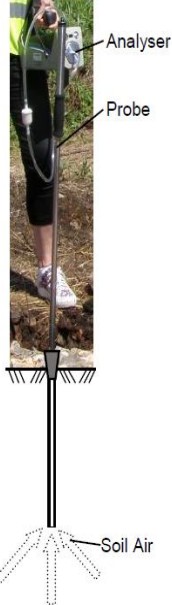

**Figure 1.** Analysis of soil air pollution.

In the environment are the most widespread light non-aqueous phase liquid LNAPL, polycyclic aromatic hydrocarbons (PAH) [18]. It is also likely that, as oil production and its transport increases, water, soil and sediment pollution will continue to increase [19,20]. In the environment, PAHs are subject to volatility, photo oxidation, chemical oxidation, and adsorption to soil particles [21]. The analyser consists of two independent analytical systems, a photoionization analyser (PID) and an infrared analyser (IR). Part of the PID analyser measures the total VOC level to ppb levels with a wide dynamic range (0.1 ppb to 4000 ppm). Part of the IR contains four independent channels for selected analyses of $CH_4$, petroleum hydrocarbons (TP), $CO_2$ with a resolution of ppm. The PID analyser has two ranges: HISENS—detection limit is low ppb values; upper limit is 100 ppm (output also in $\mu g/m^3$). HISENS mode is extremely sensitive, suitable for measuring soil air contamination-near petrol pumps etc. STANDARD—detection limit 0.1 ppm upper limit 3000 ppm (output also in mg/ $m^3$). The analyser also collects oxygen, soil temperature, temperature of the sampled gas and ambient pressure/vacuum for taking samples [17,22].

After obtaining the measured data ($CH_4$, $CO_2$, TP, PID, $O_2$), these were subsequently evaluated using statistical analyses:

1. Definition of the basic statistical characteristics of each from the indicators which were observed.
2. Verification of the dependability between the examined quantities measured during identification of soil air pollution in the monitored environmental load.
3. Through regression analysis, a mathematical model has been defined for the confirmed dependencies of the studied quantities.
4. The variability of the measured variables was analysed due to changed input quantities by a non-parametric Wilcox test [23].

The data collected were analysed using descriptive statistics, calculating the basic statistical characteristics for each indicator-average, standard deviation, minimum, maximum, amount, variation coefficient [24].

Subsequently, we monitored the indicator which has the highest or lowest means and low or high variability (standard deviation). These indicators are in some cases more difficult to compare due to different units and due to different value range (min-max), that is why the coefficient of variation (CV) was defined, which expresses the deviation of measured values from the mean expressed in percentage (Table 1).

**Table 1.** Statistical characteristics of numerical data.

| Indicator | N | DF | Mean | Std Dev | Sum | Min. | Max. | CV |
|---|---|---|---|---|---|---|---|---|
| $CH_4$ mg/m³ | 186 | 185 | 22.18 | 127.70 | 4125.3 | 0.01 | 1549.5 | 575.8% |
| $CO_2$ mg/m³ | 186 | 185 | 33,800.8 | 56,231.1 | 6,286,942.0 | 2339.2 | 600,011.0 | 166.4% |
| TP mg/m³ | 186 | 185 | 36.87 | 179.48 | 6857.0 | 0 | 1585.5 | 486.8% |
| PID mg/m³ | 186 | 185 | 5.80 | 18.32 | 1077.8 | 0 | 198.37 | 316.1% |
| $O_2$% | 186 | 185 | 16.92 | 2.02 | 3147.4 | 8.44 | 19.14 | 11.9% |
| Vapor temp. (°C) | 186 | 185 | 25.13 | 4.41 | 4673.8 | 16.67 | 36.58 | 17.5% |
| depth (m) | 186 | 185 | 3 | 1.00 | 558 | 2 | 4 | 33.4% |

Note. N—measure number; DF—records the associated degrees of freedom (DF for short) for each source of variation; Std Dev—the standard deviation; CV—coefficient of correlation; TP—hydrocarbons overall.

High CV values as it is in the case of the indicators $CH_4$, $CO_2$, TP and PID, draws attention to high fluctuation of measured values, which can in our case be connected to the influence of categorical variables—the depth of measurement and date, which was further studied by variable analysis in other sections. The low value of CV in $O_2$ according to the temperature of the vapours suggests that we are dealing with indicators with relatively stable measured values with fluctuation up to 20% compared to the mean in Table 1.

Using multidimensional analysis, we assessed the linear dependency of indicators in the JMP software environment [25,26].

We analysed all indicators with each other and examined whether there was a linear relationship between them, defined by the correlation coefficient $r$:

$$r = \frac{\sum_{i=1}^{n} (X_i - \overline{X})(Y_i - \overline{Y})}{\sqrt{\sum_{i=1}^{n}(X_i - \overline{X})^2}\sqrt{\sum_{i=1}^{n}(Y_i - \overline{Y})^2}} \qquad (1)$$

where:

$X_i$—variable $X$ observed at time $i$,
$\overline{X}$—arithmetic mean of the $X$ variables in time series,
$Y_i$—variable $Y$ observed at time $i$,
$\overline{Y}$—arithmetic mean of variables $Y$ in time series
$n$—range of time series examined

The correlation coefficient measures the two-sided linear dependence of the two variables and acquires values from the range <−1.1>, where $r = 1$ exists between variables positive linear dependence. In the case of $r = -1$, negative dependence is demonstrated. If the correlation $n = 0$, there is no dependency between $X$ and $Y$. The correlation coefficient may also have other values which can be classified as follows:

$0 < |r| < 0.3$ low level of dependency between variables,
$0.3 \leq |r| < 0.5$ moderate level of dependency between variables,
$0.5 \leq |r| < 0.7$ medium level of dependency between variables,
$0.7 \leq |r| < 1$ a strong level of dependency between variables.

For pairs of indicators with a correlation coefficient higher than $|0.5|$, we then performed a regression analysis, expressing the relationship between variables numerically.

By empirical determination of the type of analytical function and its numeric constants, we express the progress of the measured values of the dependent variable ($y$) at the changing values of the argument ($x$) of the independent variable. The regression model defines what change to variable ($y$) is caused by the change in variable $x$. However, in addition to variable $x$, variable ($y$) is influenced by other factors that we may identify as random effects. We refer to these random effects in a function as a random component of the model. Then the regression function will be expressed as follows:

$$y_i = f(x_i) + \varepsilon_i \qquad (2)$$

where

$i = 1–n$, $n$ is the number of observations,
$x_i$—the values of the explanatory variable that the experimentalist has chosen,
$y_i$—the value of the explained variable found at given values ($x_i$),

$$R^2 = 1 - \frac{\sum_{i=1}^{n}(y_i - \hat{y}_i)^2}{\sum_{i=1}^{n}(y_i - \overline{y})^2} \qquad (3)$$

$\varepsilon_i$—accidental observation error
$n$—number of observations.

We have verified the quality of the regression model through coefficient of determination ($R^2$), which takes values from interval <0.1>. $R^2$—expresses by how much percentage (0–100%) we can calculate the regression line to explain the variability of the empirical values. The closer the coefficient of determination is to (1), the more precise the regression function is.

With the usage of non-parametric analysis of variance (ANOVA) through the Wilcox test, we analysed the dispersion of values with the influence of time and depth measurement. By means of the Wilcox test, we verified a zero hypothesis (H0) at the significance level alpha = 0.05. The zero hypotheses define the equality of the achieved values of the indicator regardless of the time or measurement depth.

H0 (time effect) = There are no statistically significant differences in the values of indicator with the influence of time.

H0 (effect of the depth measurement) = There are no statistically significant differences in the values of the indicator due to the depth of measurement.

At a *p*-value of <0.05 H0 is rejected in favor of the alternative HA hypothesis. HA is defined as follows:

HA (time effect) = There are statistically significant differences in the values of the indicator with the influence of the time.

HA (impact of the depth measurement) = There are statistically significant differences in the values of the indicator due to the depth of measurement [27].

### 3. Results

In the context of the examination of linear dependency analysis, all numerical variables characterizing inputs (outdoor temperature, depth) and characterizing outputs ($CH_4$, $CO_2$, TP, PID, $O_2$) were observed. We analysed all indicators in pair, whereas we detected statistically significant positive and negative correlations.

The results of the analysis draw attention to two positive and two negative correlations, according to Table 2.

**Table 2.** Output of paired correlation indicators.

| Variable | by Variable | Correlation | Signif Prob | Positive and Negative Correlation Coefficients |
|---|---|---|---|---|
| PID ($mg/m^3$) | $CH_4$ ($mg/m^3$) | 0.1076 | 0.1438 | |
| Vapor temp. (°C) | $CO_2$ ($mg/m^3$) | 0.1126 | 0.1259 | |
| Depth (m) | $CH_4$ ($mg/m^3$) | 0.1268 | 0.0845 | |
| depth (m) | TP ($mg/m^3$) | 0.1533 | 0.0368 * | |
| Depth (m) | $CO_2$ ($mg/m^3$) | −0.1558 | 0.0337 * | |
| Vapor temp. (°C) | $O_2$ (%) | −0.1567 | 0.0327 * | |
| $O_2$ (%) | well SMV N. | 0.1643 | 0.0250 * | |
| PID ($mg/m^3$) | well SMV N. | −0.1824 | 0.0127 * | |
| Vapor temp. (°C) | well SMV N. | −0.4337 | <0001 * | |
| PID ($mg/m^3$) | TP ($mg/m^3$) | 0.4942 | <0001 * | |
| $O_2$ (%) | $CO_2$ ($mg/m^3$) | −0.6158 | <0001 * | |
| TP ($mg/m^3$) | $CH_4$ ($mg/m^3$) | 0.8127 | <0001 * | |

Signif Prob: Significance probabilities correlation; * draws attention to a statistically significant correlation.

Positive linear dependence, defined by direct proportion, was confirmed between the indicators:

- TP and $CH_4$, which has a very strong dependency with a correlation coefficient $r = 0.81$ if high TP values have been measured, then the high $CH_4$ values would be also found.
- TP and PID have mild dependency level with a correlation coefficient $r = 0.49$, this dependence define mutual growth, or decrease of monitored variables.

Both dependencies were assessed by the software as statistically significant.

Negative linear dependence, defined by inverse proportion, was confirmed among the indicators:

- for $O_2$ and $CO_2$ we are dealing with a medium strong dependence with a correlation coefficient $r = -0.6158$;
- for vapor temperature and borehole SMV N (specification of borehole (SMV), number N) there is a medium level of dependence with a correlation coefficient $r = -0.43$,

means that, with the measurement number rising, the vapor temperature decreases. This dependency is related to a measurement date which was not included in this analysis as it is recorded as a categorical variable, but in further analyses (within the frame of variability analysis), it was confirmed. That is why we have excluded the aforementioned dependency from further analysis.

Also in the case of negative dependencies, these were assessed by the software as statistically significant.

The strongest dependencies found were further analysed by regression analysis.

Based on previous findings, regression analysis was performed on pairs with the strongest linear dependency and compiled to show the mathematical expression of the relationship between variables, according to Figures 2 and 3, Tables 3 and 4.

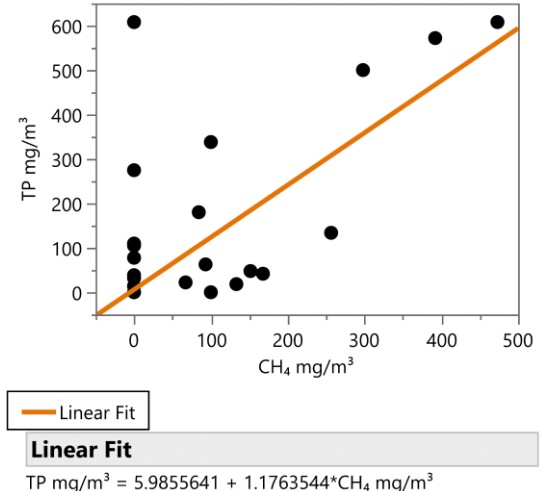

**Linear Fit**

TP mg/m$^3$ = 5.9855641 + 1.1763544*CH$_4$ mg/m$^3$

**Figure 2.** Regression analysis of the strongest dependencies of the studied parameters TP and CH$_4$.

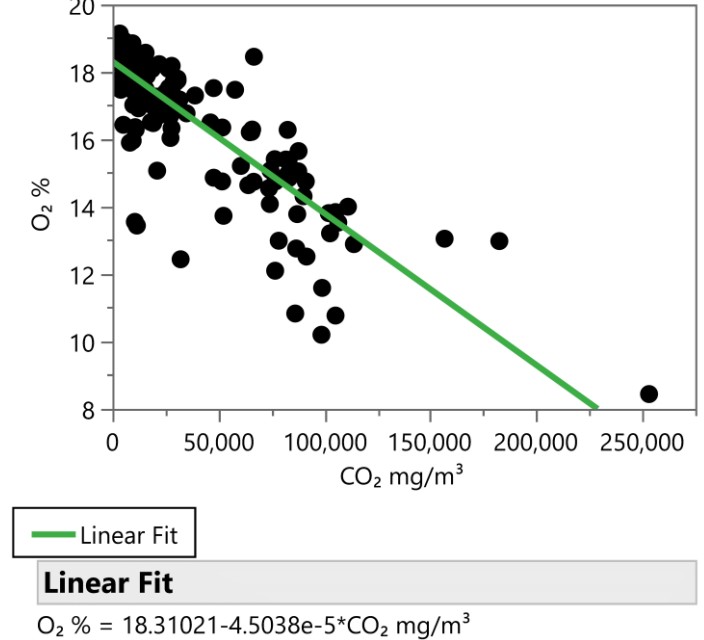

**Linear Fit**

O$_2$ % = 18.31021−4.5038e-5*CO$_2$ mg/m$^3$

**Figure 3.** Regression analysis of the strongest dependencies of the studied parameters O$_2$ and CO$_2$.

**Table 3.** Summary of Fit—Analysis of parameters TP and $CH_4$ dependencies.

| | |
|---|---|
| RSquare | 0.557431 |
| RSquare Adj | 0.554986 |
| Root Mean Square Error | 60.9382 |
| Mean of Response | 20.8876 |
| Observations (or Sum Wgts) | 183 |
| Prob > F | <0.0001 |

**Table 4.** Summary of fit—analysis of parameters $O_2$ and $CO_2$ dependencies.

| | |
|---|---|
| RSquare | 0.712529 |
| RSquare Adj | 0.710941 |
| Root Mean Square Error | 1.088111 |
| Mean of Response | 16.91656 |
| Observations (or Sum Wgts) | 183 |
| Prob > F | <0.0001 |

The results of the regression analysis are shown in Figures 2 and 3. The mathematical expression of both dependences is statistically significant and describes an important part of the unknown. In the case of TP + $CH_4$ has to do with linear dependence, when the mathematical equation describes 66% unknown. Relationship between $O_2$ and $CO_2$ is defined by a second order polynomial and describes 73% of the unknown.

Following the results of the variation coefficient in descriptive statistics presented above, it was necessary to examine the reasons for such high variability of the four variables measured—$CH_4$, $CO_2$, TP, PID. As already indicated above, two variables entered the measurement, which could have influenced this—the depth of measurement and the date. The analysis was carried out by a non-parametric Wilcox test.

The results of the Wilcox test showed that this was not a statistically significant variability of the measured values for the parameters $CH_4$, TP and PID due to the depth of measurement.

Wilcox test results showed that this was a statistically significant variability of measured values for $CO_2$ parameters according to depth and date and PID by date, visible at Figures 4–6 and Tables 5–7.

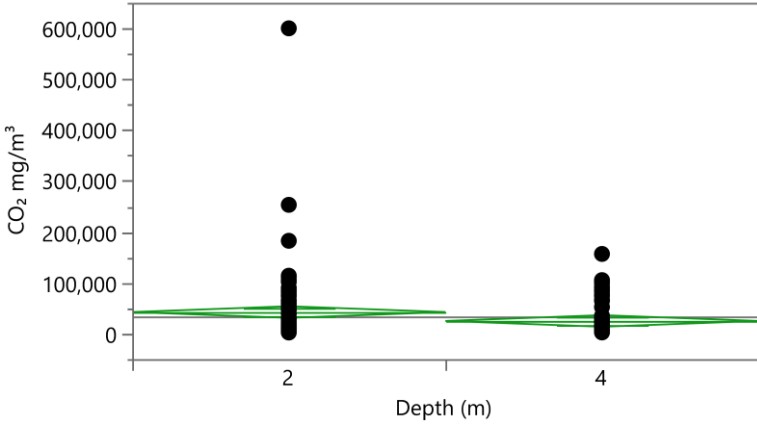

**Figure 4.** Analysis of $CO_2$ variability with the influence of the date.

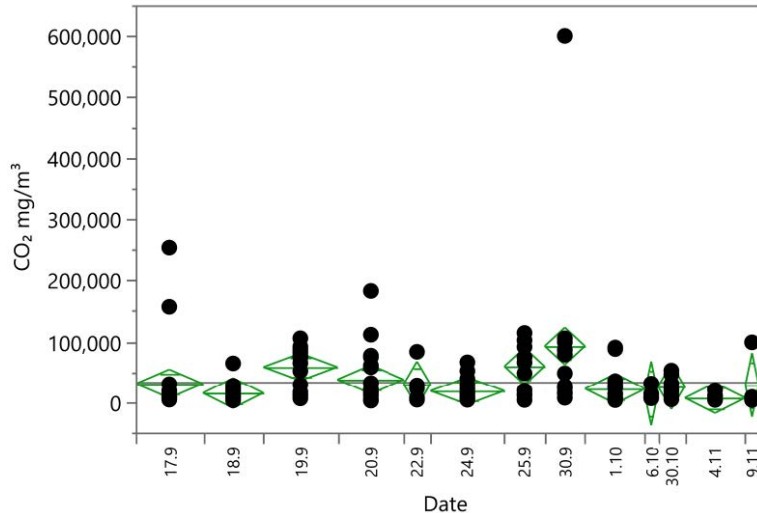

**Figure 5.** Analysis of variability $CO_2$ with the influence of the depth.

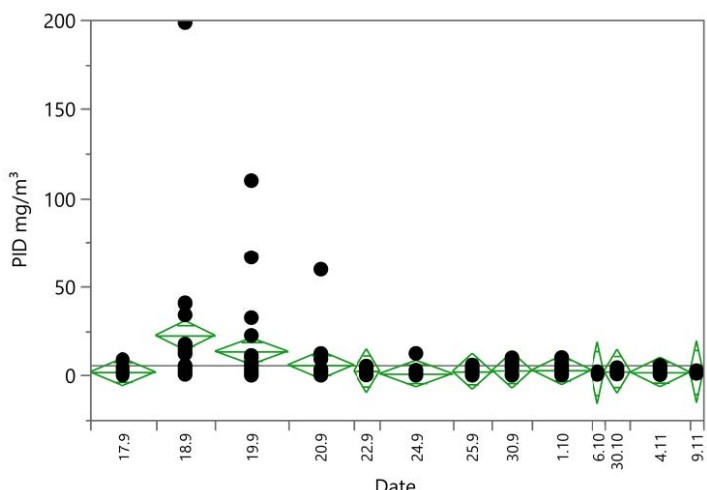

**Figure 6.** Variability analyses PID with the influence of the date.

**Table 5.** Result of Kruskal–Wallis tests—analysis of $CO_2$ variability with the influence of the date.

| Level | Count | Score Sum | Expected Score | Score Mean | (Mean-Mean0)/Std0 |
|-------|-------|-----------|----------------|------------|-------------------|
| 2 | 93 | 9467.0 | 8695.5 | 11.796 | 2.1 |
| 4 | 93 | 7924.0 | 8695.5 | 85.204 | −2.1 |

| 2-Sample Test, Normal Approximation | | |
|---|---|---|
| S | Z | Prob > \|Z\| |
| 7924.0 | −2.10011 | 0.0357 |
| 1-Way Test, ChiSquare Approximation | | |
| ChiSquare | DF | Prob > ChiSq |
| 4.4162 | 1 | 0.0356 |

**Table 6.** Result of Kruskal–Wallis tests—analysis of variability $CO_2$ with the influence of the depth.

| ChiSquare | DF | Prob > ChiSq |
|-----------|-----|--------------|
| 47.2986 | 12 | <0.0001 |

DF: Records the associated degrees of freedom (DF for short) for each source of variation.

**Table 7.** Result Kruskal–Wallis tests—analysis of variability PID with the influence of the date.

| ChiSquare | DF | Prob > ChiSq |
|---|---|---|
| 45.8965 | 12 | <0.0001 |

The results of the analysis (Figure 5 and Table 6) showed statistically significant variability in the measured $CO_2$ values, both due to the influence of the date and depth measurement. Measurements taken at a depth of two meters have higher $CO_2$ values than those taken at a depth of 4 m. Analysis of measured values according to the date highlights days when significantly higher values were measured, such as 9, 19, 25, 30 September, and days when values were measured significantly lower than the mean of measurements such as 4 November.

The results of the analysis (Figure 6 and Table 7) showed statistically significant variability in the measured PID values with the influence of the date. Analysis of measured values according to the date highlights days when significantly higher values were measured, such as 18 and 19 September and days when values were measured significantly lower than mean of measurements, for example, 24 September.

## 4. Discussion

In case of necessity, for monitoring of soil air in Slovakia and the Czech Republic, we used an in situ atmospheric geochemical measurement apparatus with the help of the analyser, or ex situ by sampling samplers.

In practice, the most commonly implemented methods are pollution surveys of old environmental burdens and accidents. Pollutants such as pesticides, fertilizers, bacteria, toxic metals and other potentially harmful (not yet identified) environmental contaminants may worsen the quality of drinking water in many regions of the world. These chemicals are carcinogenic for humans [28]. At present, however, these materials are used in industrial plants, the main advantage of which is that they save mineral resources and as a result only a small amount of $CO_2$ is released into the environment [29].

Within the framework of the survey and the remediation of polluted soils and grounds a range of geological and engineering activities are carried out, from the detection and evaluation of the current conditions of the affected areas, through the proposal and implementation of remediation and remediation measures, to post remediation monitoring and active environmental control [30,31].

Soil air monitoring is mainly used in initial screening considering its operational efficiency and speed, as well as low exploration costs. A wide range of exploratory works can be summarized as follows:

- research of archival documents and scientific literature, terrain inspection of the site realization of probes and boreholes;
- atmospheric geochemical surveys—detection of contaminants in soil air, samples of soils, rocks, sediment reservoirs, river sediments, construction structures sampling of groundwater, surface water and waste water;
- terrain measurement of hydro chemical water parameters and physical, chemical soil parameters;
- terrain and laboratory analyses of both organic and inorganic substances through long-term measurements by automatic sensors;
- hydrodynamic tests (pumping); mathematical modeling of groundwater flow, transport of substances or geochemical processes;
- evaluation of results, processing of reports and map outputs determination of the degree and type of pollution of the rock environment;
- proposal of optimal remediation measures for the disposal of pollution in the rock environment by means of in situ or ex situ development of a remediation project;
- active remediation of pollution and reclamation of affected areas (used are air sparging, bioremediation, venting, air stripping, etc.)

The atmospheric geochemical measurements in this article provide information for the determination of relative concentration of pollutants and for the determination of the extent of pollution. The measurement of volatile contaminants and petroleum substances in the soil air was carried out for this survey by the terrain analyser. The contents of $CO_2$, $CH_4$, $O_2$, PID and TP were measured. From these measurements, statistical data processing of the measured substances was developed using the JMP 15 software [25,26]. The aforementioned statistical data processing shows a clear dependence between the measured values, according to Table 2:

1.  PID and TP substances-TP and PID are related in such a way that part of the PID analyser monitors volatile VOC substances. The volatile organic components of petroleum substances, the subgroup VOC, are BTEX-aromatic hydrocarbons (benzene, toluene, ethyl benzene, and xylene), a group indicator for benzene, toluene, ethyl benzene and xylene. They are captured by part of the PID part of the analyser, but they are organic carbon compounds, so they are also found in TP according to Table 2.

2.  For $O_2$ and $CO_2$ at high $O_2$ values, lower $CO_2$ values were measured. The dependence between $O_2$ and $CO_2$ is confirmed here, as the roots of the plants contain $CO_2$ and they receive it through the leaves. The next finding was that $CO_2$ contains $O_2$ and that the amount of $CO_2$ was increasing by decreasing depth. Indirect concentration was also related to the root system of plants containing $CO_2$ and $O_2$, according to Figure 3, Table 4. The larger root system was located at the depth of two meters and drops with depth. Higher $CO_2$ levels indicate that if there is pollution at the measuring site, it is at a depth of up to 2 m.

Wilcox test results, according to Figures 4 and 5 and Tables 5 and 6, showed statistically significant variability in measured $CO_2$ values due to influence of both, date and measurement depth. Measurements taken at a depth of 2 m have higher $CO_2$ values than those taken at a depth of 4 m. The connection is seen in the depth of the root system of the plants that covered the surface, and there was no growth of trees with deep roots in the measured places, i.e., $CO_2$ with depth dropped and also due to late measurement date. When the soil temperature drops, the steams density rises and the dispersion rate is lower, this is also related to the fact that there was no organic pollution.

3.  Analysis of PID variability by date, based on the results in Figure 6, showed statistically significant variability of measured PID values due to date. However, this analysis may be insignificant if the pollution of volatile substances has not occurred at all in these wells. However, the weather can also have an impact on this change. In wetter and colder weather, the effective porosity is lower, which has a negative impact on migration and the volume of soil air for measurement.

4.  The relationship between TP (total hydrocarbons) and $CH_4$ is correct by statistical confirmation of dependency. $CH_4$ is contained in TP, i.e., when measuring hydrocarbons, total TP methane values according to Figure 2, Table 3 are also included. With increasing $CH_4$ values, TP values also increase. While detecting TP (without methane), it is necessary to subtract $CH_4$ from the measured values of TP.

Confirmation of these connections is beneficial not only during evaluation of pollution from the measured values, but also for a better understanding of the pollution migration parameters, the time evolution of pollution, the valorization of endangered objects by the spread of contamination, and the assessment of the need for and method of remediation.

Crucial is also knowledge of the relationship between $CO_2$, $CH_4$ and $O_2$ in natural bioremediation that occurs everywhere in the soil. As a theme for further research, it may be interesting to monitor the specific time evolution of organic pollution in soil non-homogeneous environments.

## 5. Conclusions

The main aim of environmental pollution research is to identify pollution in the best possible way: to describe it so well that we can get enough information to decide how

dangerous pollution is, whether it needs to be further addressed, how it will evolve and how it should be dealt with.

When measuring in situ soil air it is necessary to take into account the influence of factors below the surface, the most important of which are: the presence of methane, soil permeability, territory, surface contamination, age of contaminant, soil temperature, soil moisture and types of contaminant.

Lighter hydrocarbons are degraded faster than heavier. After some time, the spectral composition of contaminants changes dramatically, in addition, some contaminants become non-toxic, consisting of aromatic compounds, which are outside the basic spectrum of hydrocarbons. Organic matter decline and compaction are two major processes of soil degradation. Organic amendment is a current practice to compensate the loss of organic matter. There are frequent cases of the presence of fertilizers, e.g., compost, which are used to compensate for organic pollution matter in the soil [32].

The article demonstrates how to improve the evaluation of measured data based on statistical evaluation. In our report, the main subjects of the work were tools of the analyser and the statistical software JMP 15. Research, planning and implementation of the solution have been carried out to improve and accelerate the evaluation of the measured data. This article represents a solution according to a specific set of measurements, the proposed methodology and the measurement evaluation process can be used in general. In doing so, we have shown how it is possible to improve the methodology for evaluating measurements of the atmospheric geochemical survey, to identify dependencies between individual measured values and thus to contribute to easier and faster detection of soil air pollution, from which consequently it is possible its remediation and thus the protection of the environment.

For example the study that deals with an application of vapor in the remediation of soils in the city which was decontaminated from abundant tar can be found in the literature [33].

In the work [34] the current situation of soil contamination by organic pollutants and the existing soil remediation methods were introduced.

Accidental releases of hazardous waste related to the extraction, refining, and transport of oil and gas are inevitable and have to be monitored. Petroleum facilities and pipelines present environmental pollution risks, threatening both human health and ecosystems. Research has been undertaken to enhance the conventional methods for monitoring hazardous waste problems and to improve time-consuming and cost-effective ways for leak detection and remediation processes. In this study, both diffuse and imaging (hyperspectral) reflectance spectroscopy were used for detection and characterization of petroleum hydrocarbon (PHC) contamination in latosols [35].

References [36,37] attempted to illustrate the importance of using passive soil air as an innovative investigation technique in the assessment of soil and groundwater pollutions that emanates from volatile hydrocarbon activities in industrial countries.

The importance of this issue relates to the current demands for environmental protection, to which is linked detection of the environmental burdens. The possibilities for further investigation in this area of geological survey aimed at the environment will contribute to the sustainability of the development of the green economy in Slovakia and in the EU.

**Author Contributions:** Conceptualization, E.S. and A.S.; methodology, E.S. and M.T.; software, M.T. and J.N.; validation, A.S. and G.W.; formal analysis, E.S. and A.S.; investigation, E.S. and M.T.; resources, E.S. and J.N.; data curation, E.S.; writing—original draft preparation, E.S. and M.T.; writing—review and editing, G.W.; visualization, A.S. and G.W.; supervision, E.S. and A.S.; project administration, E.S.; funding acquisition, E.S. All authors have read and agreed to the published version of the manuscript.

**Funding:** This research received no external funding.

**Acknowledgments:** The article was supported by the Survey of the portable environmental burden: Project "Research of environmental burden Chemko Strazske-waste canal", SK/EZ/MI/494, 2014 and VEGA 1/0472/18 "Complex use of sophisticated instrumentation techniques in mineralurgical research".

**Conflicts of Interest:** The authors declare no conflict of interest.

## Abbreviations

| | |
|---|---|
| SVS | soil vapor survey, |
| SVE | soil vapor extraction, |
| PCBs | polychlorinated biphenyls, |
| VOCs | volatile organic compounds, |
| NPK | fertilizer, |
| $CH_4$ | methane, |
| $CO_2$ | carbon dioxide, |
| TP | hydrocarbons overall, |
| PID | volatile organic components of petroleum substances scanned by a photo ionization detector |
| BTEX | benzene, toluene, ethylbenzene, xylene |
| LNAPL | light non-aqueous phase liquid |
| POPs | persistent organic pollutants |
| PAH | polycyclic aromatic hydrocarbons |
| IR | an infrared analyser |
| CV | coefficient of variation |
| UCG | underground coal gasification |

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
