# Peer review of "Statistical Evaluation of Quantities Measured in the Detection of Soil Air Pollution of the Environmental Burden"

_applsci, doi:10.3390/app11073294_

Round 1

Reviewer 1 Report

The article is dealing with an actual topic and the suggested methodology is interesting and highly prospective. The results and discussion is based on the experimental study.

Some suggestions: 

  1. In all tables rounding of all results should be done
  2. Fig 2 contains too much details about the statistics - it should be summed up in the text and shortened
  3. Fig 3 is strange - 2 points affecting the pattern of the curve could be deleted or discussed in the text. In the legend use of "," is wrong - should be changed . The same about all other figures - statistical description should be moved elsewhere and just described in the text
  4. Conclusions are too lengthy and should be shortened

Author Response

Dear Reviewer 1.

Thank you for your comments on our article. We believe that they will contribute to improving the quality of processing of our results.

Yours sincerely,

Erika Skvarekova

Reviewer 2 Report

This paper presents the results of potentially interesting experiment. However, I can't see any significant scientific value in the paper since it only present some basic findings. The important part of scientific article such as motivation for the research, contribution to the literature, novelty etc.

The results are presented in straithforward way and there is no real "statistical model". Instead, the authors present the results of correlation, regression and Wilcox test. While the study of correlation and statistical differences between the levels of variables seem to be justified, the question arises when it come to regressions. For example, in the figure 2 you assumed that CH4 levels causes T.P. level. Do you have any justification for that? Regression analysis may be performed only when you claim that one variable explains the other one. Furthermore, the presentation of the results is not professional. You need to prepare your own tables, instead of putting print screen from statistical packages.

Author Response

Dear Reviewer 2.

Thank you for your comments on our article. We believe that they will contribute to improving the quality of processing of our results.

Yours sincerely,

Erika Skvarekova

Reviewer 3 Report

I believe the goal is worthy, but the paper falls short on providing it. It is heavy on statistics, most run from JMP software. The model needed to be presented and run for real compounds. I inferred from the abstract that this would be done for PCBs. Perhaps I missed it, but I did not see a model presented, only comparisons and correlations of the parameters. The figures seem to have missed real physicochemical and environmental relevance that needed to be parlayed with the mathematics, e.g. the unique microbial populations in the O and A horizons, versus lower layers.

Additional comments are attached. I began to correct the English, but gave up after a few pages. There are numerous omissions and misuse of terms.

Author Response

(The authors gave the same response as above.)

Round 2

Reviewer 2 Report

Dear Authors,

Thank you for your replies to my previous comments. Below I indicate some futher improvements.

Please add a separate paragraph in the introduction section in which you highlight your contribution to the scientific knowledge, clear motivation for the research and what are the differences between your work and other studies. You partially mentioned this issue in the response for my previous comments but you should write it also in the paper.

In my opinion, the paper would benefit from professional proofreading.

Author Response

Dear Reviewer 2.

Thank you for your comments on our article. We believe that these comments will positively contribute to improving the quality of processing of our results. We responded in detail to the individual objections.

 Yours sincerely,

E.Škvareková

Reviewer 3 Report

Title should be "Statistical evaluation of quantities measured in the detection of soil air pollution of the environmental burden".

I am still concerned that the linear fit in Figures 2 and 3 is appropriate given the large outlier. I would like to see more explanation and a strengthened rationale.

Author Response

Dear Reviewer 3.

Thank you for your comments on our article. We believe that these comments will positively contribute to improving the quality of processing of our results. We responded in detail to the individual objections.

Yours sincerely,

E.Škvareková
